# Physicochemical Properties and Intestinal Health Promoting Water-Insoluble Fiber Enriched Fraction Prepared from Blanched Vegetable Soybean Pod Hulls

**DOI:** 10.3390/molecules24091796

**Published:** 2019-05-09

**Authors:** Ya-Ling Huang, I-Ting Hsieh

**Affiliations:** Department of Seafood Science, National Kaohsiung University of Science and Technology, 142, Hai-Chuan Road, Nan-Tzu District, Kaohsiung 81157, Taiwan; judy333n@hotmail.com

**Keywords:** blanching, soluble dietary fiber, vegetable soybean pod hull, fecal enzyme activity, short-chain fatty acids

## Abstract

Different methods can be used to change the fiber compositions of food, and they consequently affect the physicochemical properties and physiological activities. The present study compared the effects of a blanching treatment on the physicochemical properties of water-insoluble fiber enriched fraction (WIFF) from three varieties of vegetable soybean pod hulls (tea vegetable soybean pod hull, TVSPH; black vegetable soybean pod hull, BVSPH; 305 vegetable soybean pod hulls, 305VSPH) and evaluated their effects on intestinal health in hamsters. Blanching may increase the soluble dietary fiber (SDF) content of WIFF in the 305VSPH variety by solubilizing cell wall components and releasing water-soluble sugars. Thus, the WIFF in the 305VSPH variety after blanching may be composed of cellulose and pectic substances. The WIFF of the blanched 305VSPH (B-305VSPH) variety exhibited the highest physicochemical properties, such as a water-retention capacity (11.7 g/g), oil-holding capacity (9.34 g/g), swelling property (10.8 mL/g), solubility (12.2%), and cation-exchange capacity (221 meq/kg), of the three varieties examined. The supplementation of B-305VSPH WIFF in the diet resulted in significantly (*p* < 0.05) lower cecal and fecal ammonia; activities of fecal β-d-glucosidase, β-d-glucuronidase, mucinase, and urease; as well as higher cecal total short-chain fatty acids relative to other diets. In addition, microbial analysis suggested that fecal bifidobacteria growth was enhanced by the consumption of B-305VSPH WIFF. Therefore, B-305VSPH WIFF may be applicable as a potential functional ingredient in the food industry for the improvement of intestinal health.

## 1. Introduction

Dietary fiber has attracted increasing attention in recent years with regard to its great physiological significance. The type and source of dietary fiber affects the digestive metabolism in the gastrointestinal tract, such as the composition of microbiota and enzymatic activities [1]. Dietary fiber can be degraded by intestinal bacteria, especially, soluble dietary fiber (SDF) ferments faster than insoluble dietary fiber (IDF). Fermented fibers yield further energy for bacterial growth and end products, such as the short-chain fatty acids (SCFA), namely acetate, propionate, or butyrate, that provide a beneficial effect for intestinal health [2]. The role of dietary fiber on the decreasing risk of gastrointestinal diseases depends on the physicochemical properties of the fiber, such as water retention, cation exchanges, oil-holding capacity, and fermentability [3]. In the hindgut and feces, the activities of some bacterial enzymes (e.g., β-d-glucosidase, β-d-glucuronidase, mucinase, and ureases) are associated with compromised intestinal barrier function. Since these bacterial enzymes are active components in the intestinal bacteria, which can release active metabolites, they could impair gastrointestinal function and cause increased risk of colon carcinogenesis. Many studies have shown that dietary fibers effectively decrease these bacterial enzymes and can maintain intestinal function and health [4].

Different methods can be used to change the fiber compositions and microstructures of food, and they consequently affect the physicochemical properties and physiological activities [5]. Processing steps can also enhance the physicochemical properties of dietary fiber to obtain good quality fiber. For example, blanching is widely used in the food industry to inactivate enzymes. The temperature increase of blanching breaks weak bonds between polysaccharide chains and cleaves glycosidic linkages in dietary fiber. Consequently, depolymerization of the cell walls is achieved and the composition of the dietary fiber is also modified. In terms of the dietary fiber component, blanching can covert insoluble dietary fiber (IDF) to soluble dietary fiber (SDF) due to the solubilization of IDF during thermal treatment [6]. Chantaro et al. [7] have reported that blanching can significantly increase the SDF of carrot peels. The change in the dietary fiber fraction may occur during the thermal process. In contrast, Tanongkankit et al. [8] reported that the dietary fiber composition and content of the outer leaves of cabbage were unaffected by blanching. In addition, blanching is typically a pre-drying step for the preservation of a product, as it can reduce water activity and microbial growth. This process allows control of the final quality of the product.

Vegetable by-products can be considered as a source of dietary fibers. Among the agricultural products of Taiwan, vegetable soybeans (*Glycine max* (L.) Merr.) are a principal crop that is mainly exported to Japan. After industrial processing, vegetable soybean pod hulls represent approximately 80% of the total crop weight. Each year, more than 55,000 tons of vegetable soybean pod hull have been regarded as a waste or fertilizer, which causes environmental pollution. Vegetable soybean pod hulls have an unpleasant flavor that makes it difficult to add them into food products. This agricultural by-product has received much attention as an economical source of a potential functional ingredient (e.g., dietary fiber) [9]. We hypothesized that certain varieties of fiber-rich vegetable soybean pod hull by-products, having different physicochemical properties after blanching treatment, could be further applied as a functional ingredient for industrial use.

Thus, the aim of this study was to investigate whether changes in the composition and physicochemical properties of vegetable soybean pod hull varieties were influenced by blanching. Further, their impacts on markers (such as cecal and fecal ammonia, fecal bacterial enzymes, cecal SCFA concentration, and fecal microbiota) of intestinal health in hamsters were also evaluated.

## 2. Results and Discussion

### 2.1. Proximate Analysis

The proximate compositions of the unblanched and blanched vegetable soybean pod hull varieties are shown in Table 1. All three varieties of unblanched vegetable soybean pod hulls had low contents of lipids (1.07%–1.10%) and proteins (2.44%–2.53%), which were similar to those of blanched samples (1.06%–1.09% for lipids and 2.41%–2.54% for proteins, respectively). The ash was significantly (*p* < 0.05) higher in unblanched samples than in blanched samples. The major component of all varieties of vegetable soybean pod hulls, regardless of their unblanched or blanched state, was the total dietary fiber, followed by other carbohydrates. To effectively obtain a high yield of the fiber-rich fraction, we prepared WIFF from all three vegetable soybean pod hull varieties for subsequent analysis and use. As shown in Table 2, the contents of WIFF in the unblanched sample and blanched sample were 68.4% to 72.5% and 71.2% to 74.4%, respectively. In the blanched samples, the WIFF content in 305VSPH was significantly (*p* < 0.05) higher than those of the TVSPH and BVSPH varieties. The high percentage of WIFF in 305VSPH after blanching could imply that blanched 305VSPH can be used as a potential source of dietary fiber, based on the economical and rapid extraction of WIFF. The WIFF content found in the blanched sample was higher than those reported for pressed potato fiber (54 g/100 g) [10] and kimchi by-product (62.5 g/100 g) [11]. Among the blanched samples, WIFF obtained from blanched vegetable soybean pod hulls (71.2%–74.4%) had the highest total dietary fiber. Whether the sample was unblanched or blanched, the total dietary fiber (TDF) content of WIFF from the vegetable soybean pod hull varieties was lower than that found in raw vegetable soybean pod hulls.

IDF is the major water-insoluble fiber enriched fraction among all three varieties of vegetable soybean pod hulls. Interestingly, the IDF content was significantly (*p* < 0.05) higher in unblanched samples than in blanched samples. In contrast, the SDF content of the unblanched samples was observed to be lower than that of blanched samples. Thus, we found that the blanching process reduced the IDF content and increased the SDF content in the WIFF prepared from varieties of vegetable soybean pod hulls. However, the TDF content in the WIFF of 305VSPH treated with blanching increased significantly (*p* < 0.05). This increase is attributed to the solubilization of cell wall components and the release of water-soluble sugars in the WIFF of 305VSPH during the blanching process, increasing the percentage of TDF. A comparable value of TDF (70.7 g/100 g dry weight) in WIFF was reported for fruit pomace, i.e., lime residues, processed after blanching [12].

### 2.2. Monosaccharide Composition

The monosaccharide analysis revealed that the WIFF obtained from vegetable soybean pod hull varieties with or without blanching was composed of different monomeric sugars, constituting 63.5 to 75.7 g/100 g of the dry weight (Table 3). The total sugar content of WIFF from blanched vegetable soybean pod hulls was significantly (*p* < 0.05) higher than that of unblanched samples, with the 305VSPH variety having the highest value. Among the sugars of WIFF from blanching-treated vegetable soybean pod hulls, the predominant sugars were glucose (26.4$%–28.4%), followed by uronic acid (18.2%–20.6%), xylose (16.9%–18.0%), galactose (2.73%–2.94%), and arabinose (2.40%–2.65%). These results suggested that WIFF obtained from the blanched samples possibly consisted of cellulose followed by pectic polysaccharide and hemicellulose (i.e., xyloglucan). Except for glucose, a considerable amount of all monosaccharides in the WIFF increased during the blanching process, which is attributed to the redistribution from insoluble to soluble fiber fraction. Similar results were reported by Ramos-Aguilar et al. [13], who found that a change in the monosaccharide composition of peppers occurred during blanching. Blanching increased the amount of uronic acid in the WIFF of vegetable soybean pod hulls, which can be explained by the heat-induced solubilization of the insoluble fraction. The higher uronic acid of the polysaccharide chain would have a lower degree of branching, which would further influence the physicochemical properties of the dietary fiber [14].

### 2.3. Physicochemical Properties 

The physicochemical properties of dietary fiber affect the quality of the product. Properties, such as the bulk density, water-retention capacity, oil-holding capacity, swelling property, solubility, and cation-exchange capacity, are important indices in determining whether a food additive has a potential application in the food industry. Table 4 shows that the bulk density of WIFF in blanched samples varied from 0.34 to 0.39 g/mL, which was significantly (*p* < 0.05) lower than that in unblanched samples (0.43–0.47 g/mL). It is possible that the fiber structure of WIFF in blanched samples is looser than that of unblanched samples, especially in the 305VSPH variety. Thus, the blanching process decreased the bulk density, resulting in a larger porosity of the particle surface in the vegetable soybean pod hulls. The water-retention capacity, swelling capacity, and solubility of the fiber indicates hydration properties, which are important parameters in a food system that affect, for example, the texture of a product. As shown in Table 4, the water-retention capacity of the blanched vegetable soybean pod hulls ranged from 9.62 to 11.7 g/g, which is higher than those of unblanched samples (7.77–8.53 g/g). This difference may have been due to the release of soluble substances in the WIFF of B305VSPH subjected to blanching, which would result in higher water adsorption. The water-retention capacity values of WIFF in blanched samples were higher than the results obtained from citrus seed press meals (4.79–7.76 g/g) by Karaman et al. [15]. Compared to unblanched samples, all blanched samples had a higher swelling capacity, with the B-305VSPH variety having the highest value. It is possible that the blanching process causes loosening of the dietary fiber structure and thus the exposure of more hydrophilic groups. The WIFF obtained from blanched vegetable soybean pod hulls showed higher solubility than that of unblanched samples, with the 305VSPH variety having the highest value. In terms of the water-retention capacity, swelling property, and solubility, these properties are related to the hydration of food products. A higher water-retention capacity, swelling capacity, and solubility are attributed to the presence of high soluble fiber components. The oil-holding capacity of blanched samples was higher than that of unblanched samples. A higher oil-holding capacity has been associated to a particle size reduction, which is also consistent with the results obtained of a lower bulk density (Table 4) for blanched samples [16]. In Table 4, B-305VSPH WIFF had the highest oil-holding capacity and can be applied as a food ingredient for developing a new product requiring oil-in-water/water-in-oil emulsions. The oil-holding capacity is one of the important measurements of the rheological properties of a product, for it affects the mouth-feel and texture. In general, the oil-holding capacity is relevant to the bulk density of a material. A greater surface area of a particle provides a higher binding capacity for the lipid component [17]. Blanching influenced the cation-exchange capacity of WIFF prepared from vegetable soybean pod hulls, as shown in Table 4. Compared to WIFF obtained from unblanched samples, that from blanched samples exhibited a higher (*p* < 0.05) cation-exchange capacity, ranging from 196 meq/kg for BVSPH to 221 meq/kg for 305VSPH. The results obtained for blanched samples were higher than those found in the pasteurized paste fiber concentrate of an onion by-product [17]. The extent of the cation-exchange capacity is related to the uronic acid present in the dietary fiber. The difference in the cation-exchange capacity may be correlated to the functional groups of WIFF in blanched or unblanched samples, for the cation-exchange capacity depends on the carboxyl groups and hydroxyl side chain groups in the structure of dietary fiber. Lan et al. [18] reported that fiber with a high cation-exchange capacity could help in entrapping and disintegrating lipid emulsions, leading to decreases in the absorption of lipids and cholesterol in the intestine. From the data presented in Table 3, we found a strong positive correlation (*r* = 0.96) between the uronic acid content and the cation-exchange capacity of the WIFF tested. The higher cation-exchange capacity of blanched samples may be attributed to the sugar composition of the WIFF; the 305VSPH variety had a higher content of uronic acids.

### 2.4. Animal Growth

After being fed for 30 days, all the hamsters remained healthy and active. Therefore, the hamsters adapted well to the experimental diets and environments. The body weight gain (1.40–1.52 g/d) and food intake (7.82–7.93 g/d) of the hamsters in the four dietary groups had no statistically (*p* > 0.05) significant differences. In addition, no significant differences in the weight of the small intestine (1.50–1.5 g/100 g of body weight), cecal contents (0.48–0.57 g/100 g of body weight), cecal wall (0.61–0.72 g/100 g of body weight), and colon plus rectum (1.50–1.62 g/100 g of body weight) of hamsters among the four dietary groups were observed. 

### 2.5. Cecal pH Value, Cecal Ammonia, and Fecal Ammonia 

Table 5 lists the changes in the cecal pH value, cecal ammonia, and fecal ammonia of the hamsters fed the four diets. Hamsters fed cellulose (6.61) and UB-305VSPH WIFF (6.50) diets had a significantly (*p* < 0.05) lower pH value than those fed the control diet (7.30). It was noted that consumption of WIFF obtained from the B-305VSPH variety was associated with dramatically (*p* < 0.05) lower pH values (6.34) of the cecal contents in hamsters. It was speculated that the acidic cecal pH due to feeding with WIFF of the B-305VSPH variety may have been caused by microbiota fermentation in the intestine, which produced short-chain fatty acids (SCFA) and thus a lower acidic environment in the intestine.

Approximately 70% of the N in the diet was excreted in urine and feces. It is an effective, established way to achieve increases in the full utilization of dietary N and decreases in the N in the feces to reduce NH_3_ emissions [19]. Seradj et al. [20] reported that a lower ammonia level in the colon is related to a low pH and the type of carbohydrates in the intestinal lumen. As shown in Table 5, the addition of cellulose, UB-305VSPH WIFF, and B-305VSPH WIFF into the diet significantly (*p* < 0.05) decreased the cecal ammonia content by 25.2% to 48.0%. Interactions with metabolites, such as ammonia, can damage the colonic mucosa, possibly to the detriment of the host [21]. The amount of cecal ammonia was lower in hamsters fed the B305VSPH WIFF diet, possibly due to the growth of probiotics. For example, the increased relative abundance of *Lactobacillus* and *Bifidobaterium*, which prevent potentially toxic metabolites (i.e., ammonia), was reported by Pan et al. [22] and Dowarah et al. [23]. The addition of B-305VSPH WIFF to the diet reduced the amount of ammonia in the feces of the hamsters. The amount of fecal ammonia paralleled the cecal ammonia content. In comparison with the UB-305VSPH diet, the B-305VSPH diet led to a lower cecal ammonia concentration.

### 2.6. Activities of Bacterial Enzymes in Feces

Table 6 shows the fecal bacterial enzyme activities of the hamsters fed diets with different fiber sources. Compared with those resulting from the control diet, β-d-glucosidase activities were significantly (*p* < 0.05) decreased in the hamsters fed with cellulose, UB-305VSPH WIFF, and B-305 VSPH WIFF. Especially, the B-305VSPH diet significantly (*p* < 0.05) resulted in the lowest β-d-glucosidase activity. The fecal enzyme activities were affected by the WIFF of vegetable soybean pod hulls subjected to blanching. Moreover, the β-d-glucuronidase activity in the feces of hamsters was lower with all three kinds of fiber-rich diet than with the control diet. This was more evident in hamsters fed the B-305VSPH diet, which had a reduction of 58.4%. This study showed that β-d-glucuronidase, a bacterial enzyme produced by certain intestinal bacteria, hydrolyzed β-d-glucuronides to glucuronic acid and aglycone. Increases in the activities of β-d-glucosidase and β-d-glucuronidase can cause an elevated risk of colon cancer [24]. The link between dietary components and a reduced risk of colon cancer has been established for animals fed dietary fiber [25]. These results indicated that supplementation of B-305VSPH WIFF in the diet can reduce toxic components produced by microflora and thus contribute to improvements in intestinal health.

Mucins are major glycoprotein components of the mucous. Mucins coat the surfaces of many cell types and can be secreted to form mucus gels that function as the innate defensive system in the gastrointestinal tract [26]. It was observed that, compared to the control diet, the diets containing cellulose and the other two WIFFs led to a significant (*p* < 0.05) reduction in the fecal mucinase activity (32.6%–45.7%). The lowest mucinase activity was found in hamsters fed the B-305VSPH WIFF-containing diet (Table 6). This result suggests that the consumption of B-305VSPH WIFF may protect the mucin coat in the intestinal lumen by modulating the mucinase activity. 

Table 6 shows that supplementation of cellulose and WIFF in the diets significantly (*p* < 0.05) decreased the activity of urease as compared to the control diet, and that the B-305VSPH WIFF diet gave the lowest value. From the data in Table 5, a significant high correlation (*r* = 0.98, *p* < 0.05) was observed between the fecal ammonia content and fecal urease activities in hamsters fed different diets. The results demonstrated that incorporation of a fiber-rich ingredient significantly (*p* < 0.05) decreased fecal ammonia, as evidenced by the lower urease activity in feces. 

### 2.7. SCFA Production

In Figure 1, the amount of SCFA produced was dependent on the type of fibers that fermented in the cecum of the hamsters. The difference in SCFA levels among all the groups was that cellulose was less fermented in the cecums, while B-305VSPH WIFF was degraded and utilized by the intestinal bacteria. In our study, a higher SCFA content in the cecums of hamsters fed the B-305VSPH diet was found, and this increase was expected to demonstrate the effect of fiber that is degraded by intestinal microbiota. This may have been due to the different dietary fiber composition of the WIFF in the 305VSPH variety before and after blanching, with the presence of higher fermentable fiber components (i.e., pectin substances) in the B-305VSPH diet than in the other diets. The higher SCFA levels of the hamsters fed a B-305VSPH diet were also accompanied by a more acidic environment in the cecum, as measured by the cecal pH value (Table 5). Although the compositions of SCFA in the cecal content were significantly (*p* < 0.05) different among all dietary groups, a relatively higher SCFA component was observed in acetate, followed by butyrate and propionate. Dietary fiber fermentation products produced in the cecum include acetate, propionate, and butyrate. These components have many important physiological functions in the body. Acetate is most abundantly present in the colon and is used in the liver as a substrate for regulating the synthesis of cholesterol and fatty acids. Propionate is related to decreases in lipolysis. Butyrate can be used as fuel for colonocytes [27]. Thus, the WIFF prepared from B-305VSPH treated with blanching can promote greater SCFA production in the cecum.

### 2.8. Fecal Microbiota Composition 

The composition of intestinal bacteria depends upon the type or source of carbohydrate consumed. To understand the association between the bacterial composition and dietary fiber source, we measured the bacterial community composition in the feces of hamsters fed the control, cellulose, UB-305VSPH WIFF, and B-305VSPH WIFF diets. Table 7 shows that the supplementation of fiber-rich ingredients in the diet led to significant (*p* < 0.05) elevations in the numbers of *Lactobacillus* spp. and *Bifidobacterium* spp. and reductions in the numbers of *Clostridium perfringens* and *Escherichia coli* as compared to the control diet. Interestingly, it was apparent that B-305VSPH WIFF supplementation tended to lead to significant (*p* < 0.05) changes in the number of total bacteria living in the gut. The gut microbiota had different abilities to use the dietary fiber; such differences are attributed to variations in the chemical structure and composition of the fiber matrix [25,28]. This is probably a result of the enzymes generated by intestinal bacteria, which cleave the linkages between intra-molecular and inter-molecular interactions to different extents. In parallel with its effects on bacterial activity, B-305VSPH WIFF supplementation may inhibit the growth of pathogens and promote beneficial bacterial in the gut. Grzelak-Błaszczyk [29] reported that β-d-glucuronidase activity decreased with decreasing numbers of *Clostridium perfingens* and *Escherichia coli* in the intestinal lumen. Therefore, it is speculated that adding B-305VSPH WIFF to the diet decreased the population of β-d-glucuronidase-producing *Clostridium perfingens.* Microbial fermentation of B-305VSPH WIFF may affect the gut environment through the utilization of its dietary fiber component by beneficial bacterial. Thus, B-305VSPH WIFF may be a potential prebiotic for the modulation of intestinal function and health.

### 2.9. Strengths and Limitations

Although consumption of B-305VSPH WIFF in a diet improved intestinal health by reducing fecal bacterial activities, promoting beneficial bacterial growth, and producing high amounts of SCFA in hamsters, the strengths and limitations of the methods used in relation to the animal studies and markers of intestinal health should be considered in future research. Hamsters display many features that resemble humans in terms of their physiology. In view of the advantages in lipid metabolism features, the hamster is an ideal model for studying the metabolic relationships among diets and cholesterol synthesis or intestinal microbiota. However, the hamster is not an ideal model of human intestinal health due to the differences in large bowel functioning and anatomy. Culture-based methods are quite inexpensive, but are limited due to a number of reasons (site of sampling, focus on specific microbes of interest, and limitations in many intestinal microbiota not being culturable in the first place). In addition, measuring the loss of terminal sugar resides from mucin polymers does not absolutely define the broader degradation of mucins (where protein components can also be broken down), particularly as gastric mucins do not have the same carbohydrate motifs as intestinal ones.

## 3. Materials and Methods

### 3.1. Sample Preparation and Blanching Process

The three varieties of vegetable soybean pod hulls were tea vegetable soybean pod hulls (TVSPH), black vegetable soybean pod hulls (BVSPH), and 305 vegetable soybean pod hulls (305VSPH), which were provided by Young Sun Frozen Food Co., LTD (Pingtung, Taiwan). The samples were washed with tap water and trimmed to remove spoiled parts. Then, the samples were randomly divided into two portions. One portion was conventionally blanched in a water bath of 90 °C for 1 min (1:10, *w*/*v*) and subsequently cooled down till 4 °C in an ice bath. The other portion was processed without blanching (unblanched). Both portions were dried in an air oven at 40 °C for 48 h to a constant weight and then ground to pass through a 0.5 mm size screen.

### 3.2. Proximate Composition

The proximate composition of the vegetable soybean pod hulls was determined according to the standard methods of the Association of Official Analytical Chemists (AOAC) [30]. The moisture content was determined by weighing about 1 g of sample in a pre-weighed glass dish and drying it to a constant weight at 105 °C. Protein content was calculated by multiplying the nitrogen content obtained from the Kjeldahl procedure with a conversion factor of 6.25. Fat was extracted with petroleum ether in a Soxhlet apparatus. Ash content was dried in a muffle furnace at 550 °C to a constant weight. Carbohydrate content was determined by difference: 100 − (%proteins + %fat + %water + %dietary fiber + %ash).

### 3.3. Preparation of Water-Insoluble Fiber Enriched Fraction (WIFF)

WIFF was separated from unblanched or blanched vegetable soybean pod hull varieties according to a previously described method [31]. The dried sample was mixed with distilled water at a ratio of 1:10 (*w*/*v*), and the mixture was homogenized using an Osteriser (Sunbeam-Oster, Niles, IL, USA) on the “Hi” setting for 1 min. After being filtered, the residues were washed with deionized water and 85% ethanol and then further dried at 40 °C to obtain the WIFF.

### 3.4. Measurement of Dietary Fiber Content

The insoluble dietary fiber (IDF) and soluble dietary fiber (SDF) contents were measured by the enzymatic-gravimetric method [30]. A sample was suspended in MES-Tris buffer, followed by sequential digestion of heat-stable α-amylase (95–100 °C), protease (60 °C), and amyloglucosidase (60 °C). The resulting solution was centrifuged at 1600 g for 10 min. Then, the residue was rinsed with 70% ethanol and acetone, dried in a hot-air oven at 105 °C, cooled, and weighed for IDF determination. For the SDF determination, the filtrate was precipitated by adding 4 times the volume of the filtrate of 95% ethanol. Then, the precipitate was washed and dried in a 105 °C oven. Both IDF and SDF fractions were corrected for ash, protein, and blank. Total dietary fiber was calculated as the sum of the IDF and SDF.

### 3.5. Determination of Monosaccharide Composition

The monosaccharides of the WIFF sample were analyzed according to the method of Englyst [32], with slight modifications. In brief, WIFF (20 mg) was hydrolyzed with 12 M H_2_SO_4_ at 35 °C for 60 min and then diluted to 2 M H_2_SO_4_, and the resulting mixture was boiled for 60 min. The released monosaccharides were reduced to alditols, followed by acetylation to alditol acetates. The monosaccharides were determined to be alditol acetates using a gas chromatograph (Thermo FOCUS GC series, Milan, Italy) equipped with a Quadrex 007-225 capillary column (15 m × 0.53 mm i.d.) and a flam ionization detector. Nitrogen was used as the carrier gas and the flow rate was 2 mL/min. The oven temperature was maintained initially at 100 °C for 3 min, increased to 160 °C at 4 °C/min, and then held for 5 min. The temperatures of the injector and detector were set at 270 °C and 280 °C, respectively. The monosaccharide content was calculated on a dry weight percentage basis.

### 3.6. Animal and Diet

Eight-week-old male Golden Syrian hamsters, weighing 92.9 ± 1.0 g, were obtained from the National Laboratory Animal Center of Taiwan. After one week of acclimatization, the hamsters were randomly divided into four groups of eight hamsters per group and housed in pairs at a temperature of 24 ± 1 °C and a relative humidity of 50% with 12-h light-dark cycles in a screen-bottomed stainless steel cage. The animals had free access to food and water. The study protocol was conducted according to the animal welfare guidelines and approved by the Animal Care and Use Committee of the National Kaohsiung University of Science and Technology.

The diet was prepared based on the formulation of the AIN-93M diet with slight modifications. The diet contained casein (14 g/100 g), cellulose (5 g/100 g), sucrose (10 g/100 g), corn starch (62.1 g/100 g), soybean oil (4 g/100 g), choline bitartrate (0.25 g/100 g), l-cystine (0.18 g/100 g), AIN-93M vitamin mix (1 g/100 g), and AIN-93M mineral mix (3.5 g/100 g). The animals were randomly assigned to one of four experimental diet groups, named fiber-free (control), cellulose, UB-305VSPH, and B-305VSPH. Experimental diets containing WIFF, prepared from UB-305VSPH and B-305VSPH, were formulated by replacing the cellulose in the AIN-93 diets with 5% WIFF of UB-305VSPH or B-305VSPH, respectively. Corrections were made in the quantities of casein, oil, and starch for the protein, fat, and dietary fiber present in the sample so that all diets contained the same concentrations of fat, protein, and dietary fiber. Moreover, the control diet was prepared by replacing the cellulose with corn starch. Food intake was recorded daily, and any spillage of food pellets on the bottom of the cage was carefully collected to ensure the quality of food intake measurements. Body weight was recorded every 2 days. Feces were collected daily. The unused portion of the feces was stored at −18 °C until further use. After the 30-day feeding treatment, the hamsters were fasted for 12 h and then were anesthetized with isoflurane (Halocarbon Laboratories, River Edge, NJ, USA). The small intestine, cecal wall, and colon plus rectum were collected and weighed; the cecal content was collected and stored at −80 °C for analysis.

### 3.7. Determination of Cecal pH and Ammonia

Cecal pH was determined immediately using a pH meter (Mettler Toledo Instrument Co, Ltd., Greifensee, Switzerland). The fresh cecal content was diluted 1:3 in distilled water (*w*/*w*) and then measured with a pH meter by using the method of Castillo Andrade et al. [33], with slight modifications. For cecal ammonia determination, a diluted sample was deproteinized with an equal volume of 95% ethanol. Following centrifugation at 4025 g for 10 min, the supernatant was added to a mixture of phenol and sodium pentacyanonitrosylferrate(III) dihydrate. Subsequently, a solution containing sodium hydroxide, sodium phosphate dibasic, and sodium hypochlorite was added to the mixture, which then stood for 20 min in a 37 °C water bath. The absorbance was measured spectrophotometrically at 630 nm and the results were expressed as μmol of ammonia per gram of wet cecal content.

### 3.8. Determination of Bacterial Enzyme Activities in Feces

Bacterial enzymes were extracted from the feces samples according to the method of Huang et al. [29]. A fresh fecal sample was homogenized in 0.1 M phosphate buffer and stirred thoroughly. After being centrifuged at 1006 g for 10 min, the supernatant was used for the measurement of bacterial enzyme activities. The protein concentration in the supernatant was measured using a protein assay kit (Cat. No. 500-0006, BioRad, Hercules, CA, USA).

According to the method described by Goldin and Gorbach [34], β-glucosidase activity was determined by the amount of nitrophenol from 1 mM 4-nitrophenyl β-d-glucopyranoside (No. N7006, Sigma Chemical Co., St Louis, MO, USA) and expressed as nmol of nitrophenol produced per min per mg of fecal protein. β-d-Glucuronidase activity was determined by the amount of phenolphthalein released during the hydrolysis of 0.01 M phenolphthalein-β-d-glucuronide (No. P0501, Sigma) and expressed as μmol of phenolphthalein produced per min per mg of fecal protein. Moreover, mucinase activity was measured by the amount of reducing sugar formed upon hydrolysis of porcine gastric mucin (No. M1778, Sigma) using the method of Shiau and Chang [35], and expressed as μmol of reducing sugar released per min per mg of fecal protein. Following the method of Okuda and Fujii [36] and Ling et al. [37], urease activity was measured by the rate of release of ammonia from 0.01 M urea (No. U0631, Sigma) and expressed as nmol of ammonia liberated per min per mg of fecal protein.

### 3.9. Determination of Short-Chain Fatty Acid (SCFA) Concentration

The concentrations of the main SCFAs (e.g., acetate, propionate, and butyrate) were determined according to the method of Zhu et al. [38], with slight modifications. Briefly, cecal content (0.3 g) dispersed in cold saline (0.9%, *w*/*w*) was centrifuged at 1006 g for 10 min. The supernatant was spiked with internal standard (isocaproic) and extracted with diethyl ether. An aliquot of diethyl ether layer (1 μL) was analyzed by a gas chromatograph equipped with a flame ionization detector (FID) and a packed column (GP 10% SP1200/1% H_3_PO_4_ on 80/100 chromosorb). Nitrogen was used as the carrier gas and the flow rate was 20 mL/min. The initial oven temperature of 80 °C was maintained for 3 min, and then the temperature was raised to 130 °C at a rate of 2 °C/min. The injector temperature and detector temperature were 200 °C and 250 °C, respectively. The SCFA contents were expressed as μmol per gram of wet cecal content.

### 3.10. Determination of Fecal Microbiota

A portion of fecal sample (0.1 g) was homogenized in 9 mL of an anaerobic dilution solution containing 2 g/L of gelatin, 0.0025 g/L of reaszurin solution, 0.05% cysteine, and 500 mL/L of salt solution. Aliquots of diluted homogenates (0.1 mL) were spread on each agar plate. *Escherichia coli* were enumerated using 3M™ Petrifilm™ *E. coli* count plates (PEC) with an incubation of 37 °C for 48 h. The cultures of other bacteria were performed using selective media. Plates were incubated in anaerobic jars for *Lactobacillus* spp., *Bifidobacterium* spp., and *Clostridum perfringens*. The colonies formed were counted and expressed as log colony forming units (CFU) per gram of wet weight fecal sample. *Lactobacillus* spp. were enumerated on MRS agar with incubation at 37 °C for 48 h. *Bifidobacterium* spp. were enumerated on BIM-25 agar at 37 °C for 48 h. *Clostridum perfringens* were enumerated using TSC agar containing egg yolk solution and D-cycloserine at 37 °C for 48 h.

### 3.11. Statistical Analysis

All determinations are expressed as mean ± SD. The data were analyzed by using one-way analysis of variance (ANOVA) followed by Duncan’s multiple range tests.

Statistical analysis was performed using the Statistical Analysis System and significant differences were considered at *p* < 0.05.

## 4. Conclusions

Blanching has the potential to improve the physicochemical properties of WIFF from all three kinds of vegetable soybean pod hull varieties, especially that of 305VSPH WIFF. These excellent physicochemical properties were related to the high uronic acid contents (i.e., pectic substances), suggesting that blanching increases the soluble fraction from the insoluble fraction. In addition, the supplementation of B-305VS WIFF in a diet may effectively decrease fecal enzyme activities, reduce ammonia in cecal content and feces, and modulate gut microbiota to improve intestinal health. Furthermore, B-305VSPH WIFF was easily fermented by probiotic bacteria to produce a considerable amount of SCFA in cecal contents. Thus, WIFF of blanched 305VSPH may be applicable as a potential functional ingredient in fiber-rich products.

## Figures and Tables

**Figure 1 molecules-24-01796-f001:**
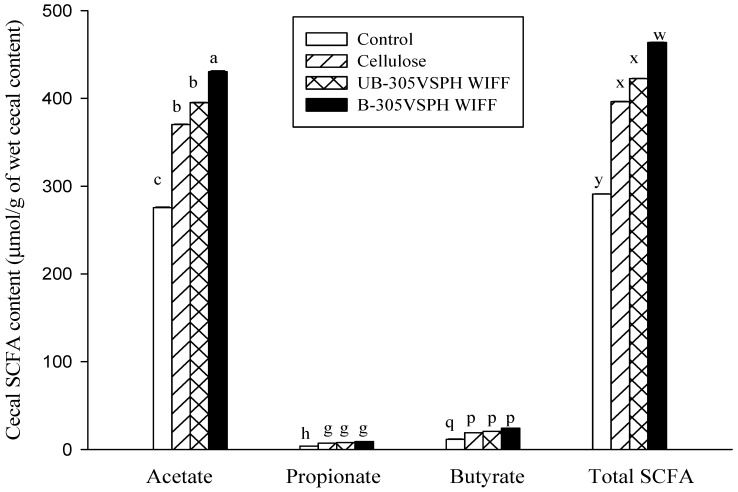
The short-chain fatty acid (SCFA) concentrations in the cecal contents of hamsters fed diets with cellulose or WIFF of UB-305VSPH and B-305VSPH. ^a-c, h-g, p-q, w-y^ Values (means ± SD, n = 8) for acetate, propionate, butyrate, and total SCFA, respectively, are significantly different (*p* < 0.05).

**Table 1 molecules-24-01796-t001:** Proximate composition of the unblanched and blanched vegetable soybean pod hull *^w^* (% dry wt.).

Sample	Protein	Lipid	TDF	Ash	Carbohydrate *^x^*
Unblanched					
UB-TVSPH	2.53 ± 0.10 *^a^*	1.10 ± 0.02 *^a^*	71.6 ± 0.02 *^c^*	1.45 ± 0.02 *^a^*	23.3 ± 0.02 *^a^*
UB-BVSPH	2.44 ± 0.08 *^a^*	1.09 ± 0.01 *^a^*	71.2 ± 0.01 *^c^*	1.48 ± 0.07 *^a^*	23.8 ± 0.01 *^a^*
UB-305VSPH	2.49 ± 0.05 *^a^*	1.07 ± 0.00 *^a^*	75.5 ± 0.02 *^b^*	1.46 ± 0.06 *^a^*	19.5 ± 0.01 *^b^*
Blanched					
B-TVSPH	2.54 ± 0.06 *^a^*	1.09 ± 0.02 *^a^*	74.9 ± 0.01 *^b^*	1.28 ± 0.05 *^a^*	20.2 ± 0.02 *^b^*
B-BVSPH	2.41 ± 0.07 *^a^*	1.07 ± 0.01 *^a^*	74.5 ± 0.04 *^b^*	1.26 ± 0.07 *^a^*	20.8 ± 0.01 *^b^*
B-305VSPH	2.45 ± 0.05 *^a^*	1.06 ± 0.04 *^a^*	79.0 ± 0.02 *^a^*	1.27 ± 0.04 *^a^*	16.2 ± 0.02 ^c^

*^a-c^* Values (means ± S.D. of triplicates) in the same column with different superscripts were significantly different at *p* < 0.05 (Duncan). UB-TVSPH = unblanched tea vegetable soybean pod hull; UB-BVSPH = unblanched black vegetable soybean pod hulls; UB-305VSPH = unblanched 305 vegetable soybean pod hull; B-TVSPH = blanched tea vegetable soybean pod hull; B-BVSPH = blanched black vegetable soybean pod hull; B-305VSPH = blanched 305 vegetable soybean pod hull; TDF = total dietary fiber. *^w^* The moisture contents of the unblanched TVSPH, BVSPH, and 305VSPH are 6.52% ± 0.01%, 6.54% ± 0.02%, and 6.53% ± 0.02%, respectively. The moisture contents of the blanched TVSPH, BVSPH, and 305VSPH are 5.74% ± 0.11%, 5.72% ± 0.04%, and 5.73% ± 0.06%, respectively. *^x^* Carbohydrate content was determined by difference: 100 − (%proteins + %fat + %water + %dietary fiber + %ash).

**Table 2 molecules-24-01796-t002:** The yield of water-insoluble fiber enriched fraction (WIFF) in unblanched and blanched vegetable soybean pod hulls and its dietary fiber content *^w^*.

Sample	WIFF Yield (% Dry wt.)	WIFF		
		IDF (% Dry wt.)	SDF (% Dry wt.)	TDF (% Dry wt.)
Unblanched				
UB-TVSPH	69.6 ± 0.21 *^c^*	62.5 ± 0.08 *^b^*	6.27 ± 0.08 *^d^*	68.77 ± 0.15 *^b^*
UB-BVSPH	68.4 ± 0.10 *^c^*	62.8 ± 0.15 *^b^*	6.06 ± 0.07 *^d^*	68.86 ± 0.12 *^b^*
UB-305VSPH	72.5 ± 0.30 *^b^*	64.9 ± 0.05 *^a^*	8.24 ± 0.08 *^c^*	73.14 ± 0.08 *^c^*
Blanched				
B-TVSPH	72.4 ± 0.02 *^b^*	57.3 ± 0.08 *^d^*	12.5 ± 0.15 *^b^*	69.8 ± 0.08 *^b^*
B-BVSPH	71.2 ± 0.10 *^b^*	57.2 ± 0.16 *^d^*	12.1 ± 0.12 *^b^*	69.3 ± 0.11 *^b^*
B-305VSPH	74.4 ± 0.32 *^a^*	58.2 ± 0.08 *^c^*	17.5 ± 0.05 *^a^*	75.7 ± 0.05 *^a^*

*^a-d^* Values (means ± S.D. of triplicates) in the same column with different superscripts were significantly different at *p* < 0.05 (Duncan). *^W^* Values are expressed as g/100 g of dry WIFF. UB-TVSPH = unblanched tea vegetable soybean pod hull; UB-BVSPH = unblanched black vegetable soybean pod hull; UB-305VSPH = unblanched 305 vegetable soybean pod hull; B-TVSPH = blanched tea vegetable soybean pod hull; B-BVSPH = blanched black vegetable soybean pod hull; B-305VSPH = blanched 305 vegetable soybean pod hull. IDF = insoluble dietary fiber; SDF = soluble dietary fiber; TDF = total dietary fiber.

**Table 3 molecules-24-01796-t003:** Monosaccharide composition *^w^* of WIFF prepared from the unblanched and blanched vegetable soybean pod hulls.

WIFF	Rhamnose	Fucose	Arabinose	Xylose	Mannose	Galactose	Glucose	Uronic Acid	Total Sugars
Unblanched									
UB-TVSPH	0.31 ± 0.01 *^e^*	Tr *^x^*	0.88 ± 0.01 *^d^*	15.2 ± 0.10 *^b^*	0.82 ± 0.06 *^d^*	1.46 ± 0.04 *^d^*	30.7 ± 0.20 *^a^*	15.4 ± 0.10 *^d^*	64.8 ± 0.01 *^c^*
UB-BVSPH	0.26 ± 0.04 *^d^*	Tr	0.86 ± 0.01 *^d^*	15.1 ± 0.20 *^b^*	0.76 ± 0.03 *^d^*	1.41 ± 0.07 *^d^*	30.5 ± 0.30 *^a^*	14.6 ± 0.04 *^d^*	63.5 ± 0.03 *^c^*
UB-305VSPH	0.49 ± 0.01 *^c^*	Tr	1.09 ± 0.01 *^c^*	16.0 ± 0.10 *^ab^*	1.24 ± 0.01 *^c^*	1.85 ± 0.02 *^c^*	32.4 ± 0.15 *^a^*	17.2 ± 0.45 *^c^*	70.3 ± 0.02 *^b^*
Blanched									
B-TVSPH	0.69 ± 0.02 *^b^*	Tr	2.43 ± 0.05 *^b^*	17.0 ± 0.10 *^a^*	1.83 ± 0.08 *^b^*	2.78 ± 0.01 *^b^*	26.6 ± 0.15 *^b^*	19.6 ± 0.07 *^b^*	70.9 ± 0.01 *^b^*
B-BVSPH	0.65 ± 0.04 *^b^*	Tr	2.40 ± 0.03 *^b^*	16.9 ± 0.21 *^a^*	1.79 ± 0.09 *^b^*	2.73 ± 0.12 *^b^*	26.4 ± 0.25 *^b^*	18.2 ± 0.08 *^bc^*	69.1 ± 0.01 *^b^*
B-305VSPH	0.81 ± 0.01 *^a^*	Tr	2.65 ± 0.02 *^a^*	18.0 ± 0.15 *^a^*	2.25 ± 0.02 *^a^*	2.94 ± 0.01 *^a^*	28.4 ± 0.15 *^b^*	20.6 ± 0.11 *^a^*	75.7 ± 0.02 *^a^*

*^a-d^* Values (mean ± S.D. of triplicates) in the same column with different superscripts are significantly different at *p* < 0.05 (Duncan). UB-TVSPH = unblanched tea vegetable soybean pod hull; UB-BVSPH = unblanched black vegetable soybean pod hull; UB-305VSPH = unblanched 305 vegetable soybean pod hull; B-TVSPH = blanched tea vegetable soybean pod hull; B-BVSPH = blanched black vegetable soybean pod hull; B-305VSPH = blanched 305 vegetable soybean pod hull. *^w^* Expressed as g/100 g dry weight. *^x^* Tr: trace amount (<0.01) (g/100 g).

**Table 4 molecules-24-01796-t004:** Physicochemical properties of WIFF prepared from unblanched and blanched vegetable soybean pod hulls.

WIFF	Bulk Density (g/mL)	Water-Retention Capacity (g/g)	Oil-Holding Capacity (g/g)	Swelling Property (mL/g)	Solubility (%)	Cation-Exchange Capacity (meq/kg)
Unblanched						
UB-TVSPH	0.46 ± 0.01 *^a^*	8.12 ± 0.01 *^d^*	5.64 ± 0.03 *^d^*	7.21 ± 0.01 *^d^*	9.29 ± 0.02 *^c^*	187 ± 0.03 *^d^*
UB-BVSPH	0.47 ± 0.01 *^a^*	7.77 ± 0.01 *^e^*	5.62 ± 0.01 *^d^*	7.02 ± 0.03 *^d^*	8.17 ± 0.01 *^d^*	175 ± 0.01 *^e^*
UB-305VSPH	0.43 ± 0.01 *^b^*	8.53 ± 0.01 *^c^*	6.73 ± 0.02 *^c^*	8.07 ± 0.03 *^c^*	10.1 ± 0.02 *^b^*	201 ± 0.09 *^b^*
Blanched						
B-TVSPH	0.38 ± 0.01 *^c^*	10.1 ± 0.01 *^b^*	8.28 ± 0.02 *^b^*	9.23 ± 0.02 *^b^*	11.1 ± 0.01 *^be^*	209 ± 0.03 *^b^*
B-BVSPH	0.39 ± 0.01 *^c^*	9.62 ± 0.01 *^b^*	8.23 ± 0.02 *^b^*	9.01 ± 0.04 *^b^*	10.5 ± 0.01 *^b^*	196 ± 0.07 *^c^*
B-305VSPH	0.34 ± 0.01 *^d^*	11.7 ± 0.01 *^a^*	9.34 ± 0.01 *^a^*	10.8 ± 0.04 *^a^*	12.2 ± 0.01 *^a^*	221 ± 0.02 *^a^*

*^a-e^* Values (means ± S.D. of triplicates) in the same column with different superscripts are significantly different at *p* < 0.05 (Duncan). UB-TVSPH = unblanched tea vegetable soybean pod hull; UB-BVSPH = unblanched black vegetable soybean pod hull; UB-305VSPH = unblanched 305 vegetable soybean pod hull; B-TVSPH = blanched tea vegetable soybean pod hull; B-BVSPH = blanched black vegetable soybean pod hull; B-305VSPH = blanched 305 vegetable soybean pod hull.

**Table 5 molecules-24-01796-t005:** Cecal pH values, cecal ammonia, and fecal ammonia of hamsters fed diets with cellulose or WIFF of the UB-305VSPH and B-305VSPH varieties.

Diet	Cecal pH	Cecal Ammonia (μmol/g cecal Content)	Fecal Ammonia (μmol/g Fresh Feces)
Control	7.30 ± 0.06 *^a^*	4.60 ± 0.06 *^a^*	49.9 ± 0.67 *^a^*
Cellulose	6.61 ± 0.02 *^b^*	3.44 ± 0.47 *^b^*	35.1 ± 0.55 *^b^*
UB-305VSPH WIFF	6.50 ± 0.06 *^b^*	2.85 ± 0.12 *^b^*	33.2 ± 0.20 *^b^*
B-305VSPH WIFF	6.34 ± 0.02 *^c^*	2.39 ± 0.03 *^c^*	28.1 ± 0.17 *^c^*

*^a-c^* Values (means ± SD, n = 8) with different letters are significantly different (*p* < 0.05). UB-TVSPH = unblanched tea vegetable soybean pod hull; UB-BVSPH = unblanched black vegetable soybean pod hull; UB-305VSPH = unblanched 305 vegetable soybean pod hull; B-TVSPH = blanched tea vegetable soybean pod hull; B-BVSPH = blanched black vegetable soybean pod hulls; B-305VSPH = blanched 305 vegetable soybean pod hull.

**Table 6 molecules-24-01796-t006:** Fecal enzyme activities of the hamsters fed diets with cellulose or WIFF of UB-305VSPH and B-305VSPH.

Diet	β-d-glucosidase (nmol Nitropheno/min/mg Protein)	β-dglucuronidase (μmol Phenolphthalein/min/mg Protein)	Mucinase (μmol Reducing Sugar/min/mg Protein)	Urease (nmol Ammonia/min/mg Protein)
Control	75.2 ± 0.57 *^a^*	2.14 ± 0.07 *^a^*	0.92 ± 0.02 *^a^*	110 ± 0.40 *^a^*
Cellulose	57.9 ± 0.44 *^b^*	1.31 ± 0.01 *^b^*	0.62 ± 0.03 *^b^*	77.5 ± 0.14 *^b^*
UB-305 VSPH WIFF	55.4 ± 0.86 *^b^*	1.25 ± 0.03 *^b^*	0.61 ± 0.05 *^b^*	74.9 ± 0.90 *^b^*
B-305VSPH WIFF	47.1 ± 0.72 *^c^*	0.89 ± 0.01 *^c^*	0.50 ± 0.02 *^c^*	68.4 ± 0.18 *^c^*

*^a-d^* Values (means ± SD, n = 8) with different letters are significantly different (*p* < 0.05). UB-TVSPH = unblanched tea vegetable soybean pod hull; UB-BVSPH = unblanched black vegetable soybean pod hull; UB-305VSPH = unblanched 305 vegetable soybean pod hull; B-TVSPH = blanched tea vegetable soybean pod hull; B-BVSPH = blanched black vegetable soybean pod hull; B-305VSPH = blanched 305 vegetable soybean pod hull.

**Table 7 molecules-24-01796-t007:** Viable bacterial counts in the fecal content of hamsters fed diets with cellulose or WIFF of UB-305VSPH and B-305VSPH.

Diets	Log CFU/g of Wet Feces			
	*Lactobacillus* spp.	*Bifidobacterium* spp.	*Clostridium* *perfringens*	*Escherichia* *coli*
Control	5.20 ± 0.01 *^c^*	5.39 ± 0.01 *^c^*	6.54 ± 0.02 *^a^*	7.39 ± 0.02 *^a^*
Cellulose	5.65 ± 0.02 *^b^*	5.84 ± 0.01 *^b^*	6.11 ± 0.01 *^b^*	7.04 ± 0.01 *^b^*
UB-305VSPH WIFF	5.79 ± 0.03 *^b^*	5.85 ± 0.03 *^b^*	6.12 ± 0.03 *b*	6.95 ± 0.03 *^b^*
B-305VSPH WIFF	6.17 ± 0.01 *^a^*	6.27 ± 0.02 *^a^*	5.84 ± 0.01 *^c^*	6.60 ± 0.03 *^c^*

*^a-d^* Values (means ± SD, n = 8) with different letters are significantly different (*p* < 0.05). UB-TVSPH = unblanched tea vegetable soybean pod hull; UB-BVSPH = unblanched black vegetable soybean pod hull; UB-305VSPH = unblanched 305 vegetable soybean pod hull; B-TVSPH = blanched tea vegetable soybean pod hull; B-BVSPH = blanched black vegetable soybean pod hull; B-305VSPH = blanched 305 vegetable soybean pod hull.

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
