# Peer review of "Physicochemical Properties and Intestinal Health Promoting Water-Insoluble Fiber Enriched Fraction Prepared from Blanched Vegetable Soybean Pod Hulls"

_molecules, 2019, doi:10.3390/molecules24091796_

Round 1
Reviewer 1 Report
Comments for authors are included in the attached file

Reviewer 2 Report
The current manuscript is well-written throughout and describes a series of experiments evaluating the bioactivity of novel fibre-rich fractions from soybean hulls. The following recommendations are aimed at improving the final version of the manuscript.
Line 75-76 - the second Aim needs reworded to more appropriately describe what the authors have carried out and to do so in a more neutral fashion. I suggest "Furthermore, their impact on markers of intestinal health in hamsters were also evaluated" or similar. The measurements taken are not absolute indices of "intestinal health" per se. Ideally statements of Aims should not predefine the major Results from the study. Within this study, you're "testing the impact of..." rather than "evaluating the beneficial effects...".
Table 2 - do the values in Table 2 for total, soluble and insoluble fibre also relate to % dry weight? This is unclear in the column headings. P
Line 183 - I think the rewording needs updated for clarity here. I suggest "... requiring [oil-in-water/water-in-oil] emulsions".
Line 244 - I think the term "growth" here may be more appropriately described as "increased relative abundance", "presence of" or similar. Please check and update if necessary.
Line 273 - mucus is mainly made up of water. Mucins could be described as the major functional component or dry-weight component however. Please revise this sentence for scientific accuracy.
Line 279 - the findings suggest that the particular WIFF fraction could reduce "the potential for mucus degradation". The direct effect of mucus degradation was not evaluated in this study. Please revise for clarity and accuracy.
Line 302 - does this statement refer to cholesterol synthesis in the liver? Please update for reader clarity.
Discussion - I think additional content should be included to consider the strengths and limitations of the Methods used in relation to the animal studies and markers of intestinal health, with inclusion of suggestions of additional approaches that could be considered in future work. The hamster is not the ideal model of human intestinal health due to the differences in large bowel function and anatomy. Culture-based assessment of faecal microbes is limited based for a number of reasons (site of sampling, focus on specific microbes of interest and limitations in many intestinal microbes not being culturable in the first place). Measuring loss of terminal sugar resides from mucin polymers does not absolutely define broader degradation of mucins (where protein components can also be broken down), particularly as gastric mucins do not have the same carbohydrate motifs as intestinal ones.
Some wider consideration of findings on monosaccahride composition of fibres is also important. In particular, this does not tell you much information about the complex, side chain structures that e.g. pectins would have, so may not absolutely predict/associate with physicochemical characteristics.
Line 364 - I suggest changing "filtrated" to "filtered".
Line 442 - update "procine" to "porcine".
Round 2
Reviewer 2 Report
The authors have responded to all queries raised by reviewers quickly and effectively. I believe that the manuscript is now of a suitable quality to be published in Molecules.